astrophysics

supernovae, massive stars, mass loss

**Author for correspondence:**
Morgan Fraser
e-mail: morgan.fraser@ucd.ie

# Supernovae and transients with circumstellar interaction

## Morgan Fraser

School of Physics, O'Brien Centre for Science North, University College Dublin, Belfield, Dublin 4, Ireland

MF, 0000-0003-2191-1674

It is 30 years since the characteristic signatures of interaction with circumstellar material (CSM) were first observed in a core-collapse supernova. Since then, CSM interaction has been observed and inferred across a range of transients, from the low-energy explosions of low-mass stars as likely electron-capture supernovae, through to the brightest superluminous supernovae. In this review, I present a brief overview of some of the interacting supernovae and transients that have been observed to date, and attempt to classify and group them together in a phenomenological framework.

## 1. Introduction and background

Wide-field synoptic surveys are discovering an ever-growing number of transients, with nearly 20 000 publicly reported in 2019 alone. Of these transients, just under 2000 were spectroscopically classified as supernovae, while the majority remained unclassified due to limited follow-up facilities. With such a large volume of transients it is unsurprising that rare, peculiar and unusual events are being found at an ever-increasing rate. Among these are many supernovae (SNe), and indeed other phenomena, which show signs of interaction with circumstellar material (CSM).

If a star explodes or erupts into a dense CSM, the faster moving ejecta will collide with slower moving circumstellar material. A forward shock is launched into the CSM, while a reverse shock moves back into the expanding ejecta. These shocks will convert some of the kinetic energy of the ejecta into radiation; as the gas at the shock front is heated to greater than $10^7$ K, this radiation will come in the form of ultraviolet and X-ray emission. The X-ray/UV flux will in turn ionize and excite gas surrounding the SN, giving rise to the narrow lines that are a classical signature of interaction. Along with this, non-thermal emission can arise from the acceleration of relativistic electrons at the shock (see [1–3] for comprehensive reviews of these processes).

**Figure 1.** (*a*) Colour composite (filters F555W, F656N, F675W) made from Hubble Space Telescope images of WR 124. (*b*) Hubble Space Telescope colour composite for $\eta$ Car, (filters F280N, F336W, F658N). Both WR 124 and $\eta$ Car are examples of evolved massive stars with significant CSM; 1−2 M$_\odot$ in the case of the former [6], and from approximately 10 to greater than 40 M$_\odot$ for the latter [7,8]. Image credit: NASA/ESA/Schmidt.

## 1.1. Mass loss in massive stars

Interacting supernovae and transients give an important insight into mass loss from massive stars in the final stages of their evolution. The evolution and ultimate fate of a massive (greater than 8 M$_\odot$) star is, to a large extent, governed by the mass loss it will experience over its lifetime. For a single star which is born at close to solar metallicity with a zero age main sequence (ZAMS) mass of $\lesssim 30$ M$_\odot$, stellar winds will be insufficient to remove the H envelope. If such a star explodes as a core-collapse supernova it will appear as a H-rich Type II SN [4]. For stars more massive than this, stellar winds (at solar metallicity) are sufficient to remove all or part of the H or He envelope before the star explodes [5] (figure 1). In this case, the resulting supernova will be a Type Ib or Ic event, depending on whether only the H, or both the H and He envelopes are removed. *In many cases the mass of a star at the point of explosion will be less than half of its ZAMS mass.*

The stellar winds responsible for removing this mass are ultimately driven by the transfer of momentum from photons to the gas in the outer layers of the star. While it is generally accepted that this process will scale with both the luminosity of the star and its metallicity (as UV absorption lines are required to absorb photons), many of the details remain uncertain (see [9] for a comprehensive review). In particular, recent work has shown that winds are not smooth, but are instead clumpy, with local over- and under-densities of material that will affect how radiation is absorbed. Critically, clumping can change mass loss rates by an order of magnitude or more [10,11]. The choice of mass loss rate will have a strong effect on stellar models, in particular for stars greater than 30 M$_\odot$ [12].

Along with winds, stars can also lose mass through episodic eruptions. While the physical mechanism behind these eruptions is presently unknown, there are clear historical examples of them occurring, most famously for $\eta$ Carinae. In the middle of the nineteenth century, $\eta$ Car experienced a decade-long outburst which saw it eject approximately 10 M$_\odot$, forming the spectacular Homunculus Nebula which can be observed today (figure 1*b*) [7]. The so-called 'Great Eruption' saw $\eta$ Car for a period become one of the brightest stars in the sky. The inferred mass loss rate of $\eta$ Car during this period (around 1 M$_\odot$ yr$^{-1}$) is much greater than can be accounted for by normal mass loss mechanisms, and requires either a super-Eddington wind or some kind of explosive outburst.

One of the many remarkable properties of $\eta$ Car is that some of the material lost during the Great Eruption had velocities of approximately $10^4$ km s$^{-1}$ [13]. This is much faster than the typical wind velocity of even a Wolf–Rayet (WR) star, and points towards an explosive eruption. Several of the proposed explanations for what triggered this eruption invoke mass transfer in an eccentric binary or a stellar merger [14–17], motivated in part by the coincidence between the start of one of the subsequent outbursts of $\eta$ Car and periastron [18]. Alternatively, instabilities in the later stages of nuclear burning, or perhaps associated with shell ignition could also provide the necessary energy. While many of these scenarios are plausible, the identification of ejecta at considerable distance from $\eta$ Car consistent with outbursts as long as approximately 800 years ago [19] pose a challenge.

Binarity can also be an important factor in the mass loss history of many stars. The majority of massive stars have a binary companion [20], while 70% of massive stars will exchange mass with a companion [21]. A clear example of the role a binary companion can play in removing mass from a supernova progenitor is the Type IIb SN 1993J. The progenitor candidate identified in archival images was found to have a blue excess consistent with the presence of a second source [22]. While the progenitor candidate had disappeared after SN 1993J faded, the blue source remained, and was proposed to be the surviving binary companion [23,24]. Detailed models of interacting binaries with mass transfer and stripping were able to reproduce the position of the progenitor and its companion on the Hertzsprung–Russell diagram at the point of explosion [25].

Modelling of interacting binaries can be difficult due to the often poorly constrained mass transfer efficiency [26], as well as the challenges of dynamical processes such as mergers and common envelope ejections. Despite this, progress continues to be made on pre-supernova binaries both through detailed modelling and population synthesis [27–32], while recent versions of the MESA stellar evolutionary code include mass transfer in binaries [33].

## 1.2. Interacting transients

The first suggestion that SNe with narrow lines in their spectra were a distinct class was made by Schlegel in 1990, who also proposed the 'IIn' nomenclature [34].[1] Since then, the number of Type IIn SNe with reasonably comprehensive follow-up observations has risen, with probably around 100 well-studied events in the literature to date. In this review, I present an overview of the current landscape for transients which show some evidence for circumstellar interaction, with a particular focus on supernovae.[2] Such a survey will necessarily be biased and incomplete, not least because many of the possible classes are a matter of ongoing debate in the literature. I will also largely focus on circumstellar interaction that occurs relatively soon (i.e. less than a decade) after the SN explosion. There are a number of SNe which are now more than a few decades after explosion and that are continuing to be observed [38,39] (including SN 1980K, where one of the first detections of narrow Hα was reported [40]). While such objects can provide insight into mass loss (as can studies of Galactic supernova remnants), I do not discuss them in this review for reasons of brevity, and as many of these SNe have sparse early time data making it difficult to connect the properties of the CSM to those of the SN. The reader is referred to the review of Milisavljevic & Fesen [41], which provides an excellent overview of the liminal phase between SN and remnant.

# 2. Observational classes of interacting SNe

The field of SN classification and taxonomy is rapidly evolving as new classes of transients are identified. A clear illustration of this can be seen in the review of Type Ia SNe by Taubenberger, where the prototypes of many subclasses have only been identified in the last 15 years [42].

Compounding the confusion in transient classes is that the taxonomic classification of SNe proceeds on an observational rather than physical basis. This is particularly challenging for interacting SNe, where CSM interaction may mask the underlying transient (for example, the debate over so-called 'Type Ia-CSM' SNe has shown how a CSM-enshrouded thermonuclear explosion of a white dwarf can look remarkably similar to the core collapse of a stripped star [43,44]).

However, while the spectroscopic properties of Type IIn SNe often appear similar, they can display heterogeneous light curves (see, for example, the sample of spectroscopically confirmed SNe IIn with light curves from PTF [45]). Light curves can provide a better indicator of total energetics in an interacting supernova (especially as fast ejecta is usually masked behind CSM), while the shape and duration of the light curve can provide some indication of the mass and radial distribution of CSM.

## 2.1. Luminous, long-lived Type IIn SNe

A number of Type IIn SNe display slow evolving, long-lived light curves [46,47], and are commonly believed to be associated with the explosion of a very massive star into a massive (on the order of a

---

[1]Although see Zwicky's peculiar 'Type V' SN 1961V as an example of a prior transient that is clearly affected by CSM interaction [35,36].

[2]See [37] for another recent review of this topic.

few, to a few tens of $M_\odot$) CSM. The archetype of this group is SN 2010jl, where combined X-ray, radio and optical observations allowed for detailed modelling of the CSM density [48–50]. Various authors set a lower limit to the total CSM mass of either 3 or $10\,M_\odot$, with the precise value depending to a large extent on the assumed wind velocity. Whatever the precise value, the radius of the CSM (approx. $10^{16}$ cm [48]) is consistent with a relatively recent episode of enhanced mass loss, starting around 40 years prior to the explosion. It is difficult to imagine anything other than a massive, luminous blue variable (LBV)-like progenitor (perhaps even an extra-galactic $\eta$ Car analogue) giving rise to such a CSM configuration and SN. This would also be consistent with the location of SN 2010jl in a UV-bright highly star-forming environment.

Unfortunately SN 2010jl was first discovered when it was emerging from behind the Sun, and so there are only lower limits to the rise time to peak luminosity. However, a number of other long-lived and energetic Type IIn SNe (for example, PTF12glz [51] and HSC16aayt [52]) have shown a very slow and long rise time. This could be interpreted as being indicative of a long diffusion time for photons to escape from a massive optically thick CSM. Alternatively the long rise time could be due to an asymmetric CSM [51], possibly in the form of a circumstellar disc [53]. This latter scenario would be consistent with spectropolarimetry of SN 2010jl, which revealed substantial asymmetry [54].

## 2.2. SN 2009ip-like events

One of the most intensively studied (and contentious) interacting transients is SN 2009ip. First discovered in 2009 and assigned a 'SN' designation, SN 2009ip went on to display 3 years of erratic variability. During this phase the spectrum of SN 2009ip was hot and dominated by narrow CSM lines, reminiscent of an LBV eruption. Remarkably, even during this period, spectra revealed some material moving at over approximately $12\,000$ km s$^{-1}$ (from the blue edge of the H$\alpha$ absorption) [55,56]. Such velocities are indicative of an energetic and explosive event, and an associated blast wave which was even faster than that seen in $\eta$ Car [57].

The erratic variability of SN 2009ip reached a dramatic conclusion in 2012, when the transient first brightened to around $M_R$ approximately $-15$ over one month ('event A'), before reaching $M_R$ approximately $-18$ in a second brightening event ('event B'). As no non-terminal eruption of a massive star had hitherto been observed to reach this luminosity, it was suggested that the 2012 B event was the final explosion of SN 2009ip as a (genuine) supernova [58]. Nonetheless, some authors have argued that it may not be a terminal core collapse or remained agnostic (e.g. [56,59,60]), on the basis of the restrictive upper limit to the ejected $^{56}$Ni mass and the absence of spectral features associated with SN nucleosynthesis.

If the 'event B' of SN 2009ip was indeed a genuine SN, then we can consider the implications of the pre-SN outbursts for stellar evolutionary models. Any outburst immediately prior to core collapse is plausibly triggered by one of the later phases of nuclear burning. The timescale for core Si burning is on the order of a day, while even in a $60\,M_\odot$ star core C burning commences about 50 years before core collapse [61]. Intermediate between these burning stages are core O and Ne ignition. Thöne *et al.* [62] suggested that the progenitor of SN 2015bh (another member of the SN 2009ip-like class) had been in a state of variability for at least two decades before its 2015 outburst. It is important to note that these timescales come from one-dimensional models of stellar evolution. While these models well reproduce observables such as the locus of massive stars on a Hertzsprung–Russell diagram, they are necessarily less tested when it comes to the final stages of stellar evolution. Furthermore, there is a possibility that convective or turbulent instabilities that are seen in three-dimensional models can drive outbursts [63,64]

One of the most intriguing aspects of SN 2009ip is that there are a surprising number of similar transients that have been found in the last few years [62,65–68]. These transients are almost identical in their spectra, have a similar bright peak and decline in their light curve, and share a pre-peak outburst. What makes this surprising is that SN 2009ip is a complex transient with a highly structured CSM. Multiple emission and absorption components are needed to fit the line profiles in SN 2009ip (e.g. [69]) pointing towards a complex geometry. Further evidence for this comes from spectropolarimetry [70,71], where the polarization was interpreted as implying a disc-like geometry. From this, one would naturally expect an observationally diverse set of transients, thanks to differences in CSM, mass loss history, and critically, viewing angle effects.

The true nature of the SN 2009ip-like transients remains uncertain. The steady decline and lack of new outbursts since 2012 points towards a terminal event. On the other hand, the transient has not yet faded significantly below the progenitor magnitude. Whether these events are core-collapse supernovae, extreme non-terminal eruptions, or even a star entering the Wolf–Rayet phase is an open question [62].

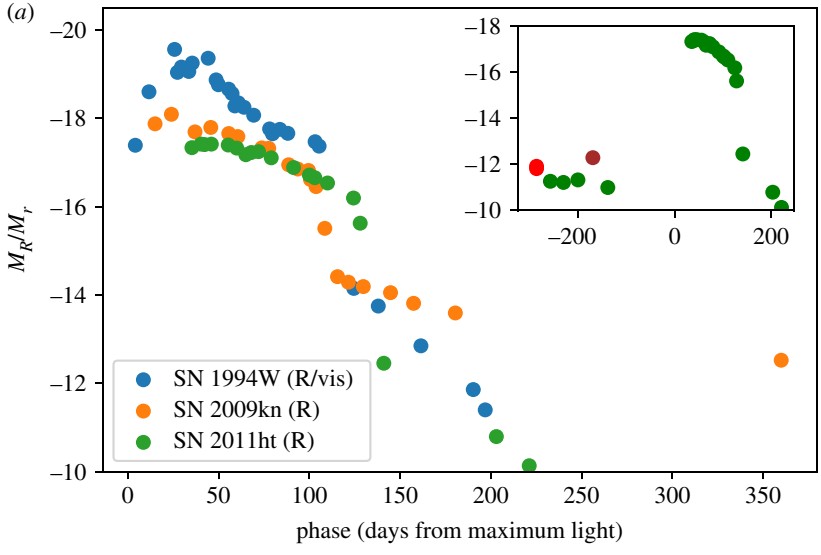

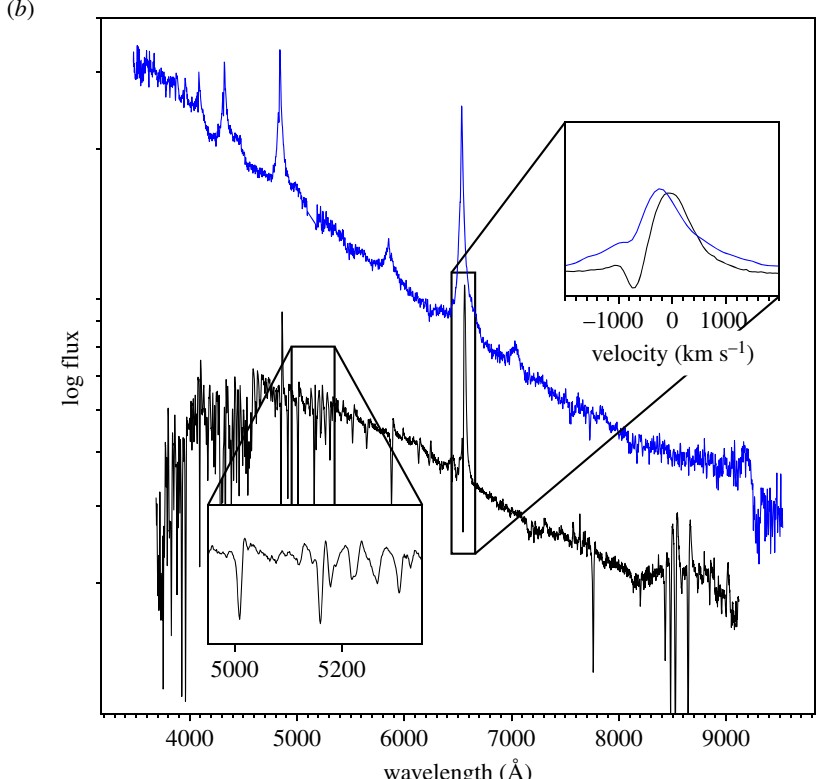

**Figure 2.** (a) Light curves for SNe IIn-P: SN 1994W [73] SN 2011ht [72,74] SN 2009kn [75]. The inset shows the precursor outburst to SN 2011ht (axes are the same as in the main plot; r, z and y-band detections are plotted in green, red and brown respectively). (b) Spectra of SN 2011ht at +54 (blue) and +116 days (black) from discovery (i.e. in the middle and at the end of the plateau phase) [76,77]. The inset panels show a zoom in on some of the weak metallic P-Cygni lines seen during the plateau, as well as the velocity profile of Hα.

## 2.3. IIn-P

The term 'IIn-P' SNe was first suggested by Mauerhan *et al.* [72], who grouped together SNe 1994W, 2009kn and 2011ht. These events all follow a similar photometric evolution, with a relatively luminous plateau lasting around one hundred days, followed by a precipitous drop onto a slowly declining tail phase (figure 2). This evolution is at least superficially similar to that seen in Type IIP supernovae.

The ejected Ni mass for this class tends to be low, with estimates for SN 1994W ranging from approximately $10^{-2}$ to $10^{-3}\,\mathrm{M_\odot}$ [73]; while SN 2009kn had a ejected Ni mass of approximately

0.02 $M_\odot$ [75]. As noted by [73], attempts to estimate or even place limits on the Ni mass for these SNe are somewhat fraught. A late-time near infrared (NIR) excess (as seen in SN 2009kn) is suggestive of dust formation, implying that we may underestimate the tail luminosity. Equally, the rapid decline seen in SN 1994W, which is faster than the 0.98 mag 100d$^{-1}$ decline rate expected from $^{56}$Ni, implies that $\gamma$-rays are not fully trapped in the ejecta, and hence some of the energy from radioactive decay may be leaking out unobserved. On the other hand, spectral signatures of CSM interaction persist to these late phases. If the SN is not entirely powered by radioactive decay, measurements of the tail phase luminosity will overestimate the Ni mass. These caveats notwithstanding, unless some material is lost through fallback onto the compact remnant, low ejected Ni masses are broadly consistent with a low-energy explosion of a star at the lower extremum of the mass range for core collapse [72,78].

The spectra of IIn-P are characterized at early times by a hot, blue continuum with narrow Balmer emission lines. The Balmer lines also display a P-Cygni absorption component with a velocity minimum around 400–800 km s$^{-1}$ [75,76]. As the spectrum cools, a forest of narrow P-Cygni Fe$_{\rm II}$ and Ti$_{\rm II}$ lines are revealed, together with both forbidden and permitted transitions of Ca. Broad (approx. $10^4$ km s$^{-1}$) ejecta features are not seen at any phase, and while [O$_{\rm I}$] lines are observed in SN 2011ht at late phases [72], these are extremely weak.

Interestingly, a precursor outburst was seen around six months prior to SN 2011ht [74]. While this is plausibly associated with the ejection of material from the progenitor, it does not allow us to infer whether SN 2011ht itself was a terminal or non-terminal event. In the case of a core-collapse SN, the precursor outburst may be driven by instabilities in the later stages of nuclear burning [74]. Similar outbursts cannot be ruled out for SNe 1994W and 2009kn, as they lack pre-discovery imaging.

While most authors have argued that SN 2011ht was a core-collapse explosion [72,75,79], it is worth noting that some have suggested that SN 1994W and SN 2011ht were actually non-terminal outbursts [76,80]. Humphreys *et al.* [76] argued that a non-terminal event could equally explain the observations of 2011ht, and that while a mass loss rate of approximately 0.05 $M_\odot$ yr$^{-1}$ would be required, this is potentially consistent with a super-Eddington giant eruption from a massive star. In the case of SN 1994W, Dessart *et al.* [80] proposed a scenario where successive shells of material ejected by the progenitor could collide, leading to a luminous transient. An alternative suggestion, based in large part on a strong spectral similarity between SN 2011ht and the luminous red nova NGC 4490-2011OT1, was that the outburst six months prior to SN 2011ht was in fact the ejection of a common envelope following a binary merger [81]. In this scenario, either the merger could have triggered a supernova explosion, or alternatively, 2011ht was caused by the collision of ejected shells.

## 2.4. SN 1998S-like events, SNe IIL and early interaction in Type IIP SNe

Non-interacting H-rich core collapses with a linearly declining light curve are designated 'Type IIL' events [82]. While there is some debate as to whether Type IIL SNe are a discrete class, or whether they form a continuum with more common Type II Plateau (IIP) events, they are generally accepted to arise from the explosion of a star that has too little H in its envelope to sustain a recombination powered plateau, but which are not so stripped as to give rise to a Type IIb SN. Interestingly, there are a number of apparent connections between Type IIL SNe and Type IIn events.

On the basis of their light curves alone, a number of purported Type IIn SNe such as SN 1998S [83] could be classified as Type IIL events. This SN experienced a linear decline, while the signatures of interaction, viz. strong narrow emission lines, only persisted for the first month after explosion [84]. It is worth noting that if SN 1998S had not been spectroscopically classified until approximately one month after explosion it would have been regarded as a Type IIL SN. A handful of similar events such as SNe 2013fc and 1996al have been found to have a similar photometric evolution [85,86].

Even among unambiguous Type IIL SNe that show no obvious signs of narrow lines, it has been suggested that CSM interaction may still provide part of the luminosity (e.g. [87]). In general, radiation hydrodynamic models struggle to reproduce a bright linear light curve with no large drop onto a radioactive tail phase. One solution to this is to invoke an extremely inflated envelope (or superwind) for the progenitor, as suggested for the Type IIL SN 1979C [88]. Similarly, more recent modelling has shown that the light curves of Type IIL SNe can be best reproduced with an additional dense CSM that is confined close to the progenitor [89,90] (the density further out would have to be sufficiently low that narrow lines are not formed [87]). Furthermore, even some apparently normal Type IIP SNe have been found to show signs of CSM interaction at early times, requiring some additional material close to the progenitor lost just before explosion [91–93]. It is worth noting that X-ray observations at late times point towards a relatively high mass loss rate ($10^{-4}$ $M_\odot$ yr$^{-1}$) for the

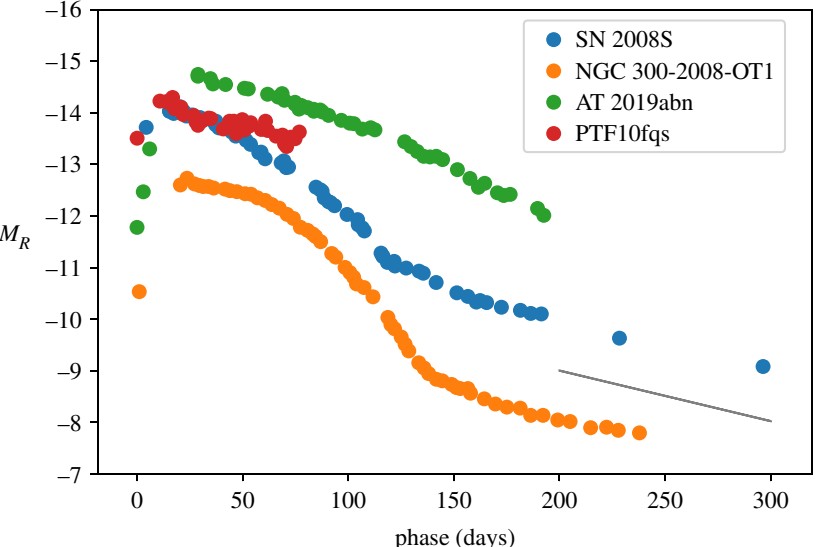

**Figure 3.** Light curves for SN 2008S-like transients [95–98]. The black line shows the expected decline rate from the decay of $^{56}$Co.

progenitor of SN 1979C [38], while fluctuations in the radio light curve are suggestive of an inhomogeneous CSM [94].

## 2.5. SN 2008S-like

The class of SN 2008S-like IIn (also commonly referred to as intermediate-luminosity red transients, ILRTs) are characterized by faint absolute magnitudes ($-12.5 > M_R > -15$), a smoothly evolving light curve that in some cases settles onto a faint tail, and a small ejected Ni mass (figure 3). Their spectra generally become quite red, and are dominated by narrow emission lines of H, Ca and Fe. In particular, the [Ca II] feature at $\lambda\lambda$ 7291, 7323 is a characteristic signature of ILRTs.[3]

The prototype of this class was SN 2008S [95], which was suggested to result from a weak electron capture supernova [101] in a super-asymptotic giant branch progenitor. The progenitor was enshrouded in a dusty circumstellar medium, as it was detected at mid-infrared wavelengths with the Spitzer Space Telescope, but not at optical or NIR wavelengths. Similar conclusions have been drawn for other members of the class [102].

Alternative explanations for some of the SN 2008S-like events as non-terminal outbursts have also been proposed, in particular that some of these may be evolved stars moving blue-ward on the Hertzsprung–Russell diagram, or a high-mass X-ray binary [103,104]. A key test for this is whether there is a surviving star after the transient has faded, which remains a point of some debate in the literature [104,105].

## 2.6. Superluminous and peculiar interacting SNe

Recent years have seen the discovery of increasing numbers of superluminous supernovae (SLSNe), with peak luminosities far in excess of what can plausibly be powered by radioactive decay [106]. Many of these SNe are H-poor, and are probably powered by some sort of additional central engine, such as a rapidly rotating magnetar which injects energy as it spins down [107]. A handful of these superluminous events are spectroscopically Type IIn SNe, where CSM interaction is presumably sufficient to boost the luminosity to brighter than $M_V$ approximately $-22$. One such example, CSS121015:004244+132827, is one of the most luminous SNe ever seen [108].

The prototype of this class is SN 2006gy, which reached a peak magnitude of $-22$, while displaying relatively narrow lines emission lines of H [109,110]. Modelling of the light curves and spectra are consistent with the collision of approximately $10\,M_\odot$ of ejecta with a similar mass of patchy CSM [110,111]. While these values are large, they are not inconceivable, and the fact that SN 2006gy was so

[3]Although some objects continue to blur the apparent boundaries between ILRTs and other classes [99,100].

luminous is simply a reminder that the conversion of kinetic energy to radiation at a shock can be a highly efficient process.

Late-time observations of H-poor SLSNe have revealed that some show strong spectroscopic signatures of interaction with H-*rich* CSM at late times (more than six months from explosion). What is remarkable is that the velocity of the Balmer emission lines that appear at such late times are approximately 5000 km s$^{-1}$. These velocities are much higher than the fastest stellar winds, and even the velocity of the bulk of the ejecta from typical LBV giant eruptions. In fact, the most plausible mechanism to eject a massive H envelope with this velocity is a pulsational pair instability explosion [112]. Additional support for this scenario comes from observations of the SLSN iPTF16eh, where a light echo seen in spectra is consistent with a massive circumstellar shell ejected 30 years before the SN [113].

Interaction has also been proposed as a power source for some peculiar H-rich SNe. While iPTF14hls only reached an absolute magnitude $r$ approximately −19 at its peak, it remained bright for over 2 years, with multiple peaks in its light curve [114,115]. Remarkably, iPTF14hls displayed broad approximately 6000 km s$^{-1}$ H lines for the same duration, with very little evolution in line strength or velocity. Arcavi *et al.* suggested that the line-forming region was detached from the photosphere, and could be coming from a massive shell of some tens of M$_\odot$ that was ejected by the progenitor a few years prior. This scenario was further strengthened by the tentative detection of a precursor outburst on photographic plates from the 1950s [114]. Remarkably, [116] presented a spectrum at greater than 3 years from discovery which showed H$\alpha$ to have a double-peaked structure, with two emission components at −1000 and +700 km s$^{-1}$. Similar multi-component H$\alpha$ emission has been seen in a number of Type IIn SNe at late phases [117,118]. Andrews & Smith [116] suggested a scenario where CSM was present in a disc around iPTF14hls, but that interaction was masked by fast ejecta that expanded out past the disc, blocking spectroscopic signatures of narrow lines for some viewing angles.

## 2.7. Ibn

While the majority of interacting transients arise from the collision of ejecta with H-rich CSM, in a handful of cases the CSM appears to be He-rich. The archetype of this class is SN 2006jc [119,120], although there were some earlier identifications of narrow He lines in SN spectra, such as SN 1999cq [121]. Since then, about 30 so-called 'Ibn' SNe have been identified.

Photometrically, Type Ibn SNe are typically characterized by a rapid decline from a relatively bright peak, consistent with a small ejecta mass. Their spectra, however, tend to be more varied [122], and in particular, there appears to be a continuum of H and He line strengths, where some events have much stronger He lines, while others appear to be intermediate between Types Ibn and IIn [123].

The progenitors of Type Ibn SNe have been suggested to be Wolf–Rayet stars (e.g. [124,125]), and this is appealing for a number of reasons. In particular, Wolf–Rayet stars are H-poor; while they have wind velocities comparable to the approximately $10^3$ km s$^{-1}$ seen in the narrow lines of Type Ibn SNe. However, it is worth noting that the fast decline of Type Ibn light curves in some cases points to a smaller ejecta mass than that of normal Type Ibc SNe. This is surprising, as the majority of the latter are believed to come from stars that have been stripped in lower-mass binaries (e.g. [126,127]). If this is the case, the progenitors of most Type Ibc SNe should hence have a smaller core mass than that of a single massive WR star at the point of explosion, and one would expect Type Ibn SNe to have *broader* light curves than Type Ibc SNe.

At least one Type Ibn SN displayed a pre-SN outburst around 2 years prior to explosion [119]. The nature of this outburst is still debated, although as SN 2006jc was probably in a binary system [128], it may well come from an LBV-like companion rather than the progenitor itself.

## 2.8. Stripped envelope SNe with late-time interaction

Stripped envelope SNe come from stars that have lost some fraction (or potentially all) of their H, or H and He envelopes. Type IIb SNe are thought to come from stars that have only a thin H envelope when they explode, as they show broad H features in their spectra which disappear after a few weeks. Type Ib SNe show no broad H in their spectra at any phase, while Type Ic SNe show neither H nor He [129].

Modelling has suggested that Type IIb SNe may come from the explosion of stars that have had much of their H envelope stripped by a binary companion [25,29,32]. While strong optical signatures for CSM interaction have not been detected at early times in Type IIb SNe (although see [130]), radio observations have pointed to the presence of CSM around some Type IIb SNe [131]. Binary progenitors are

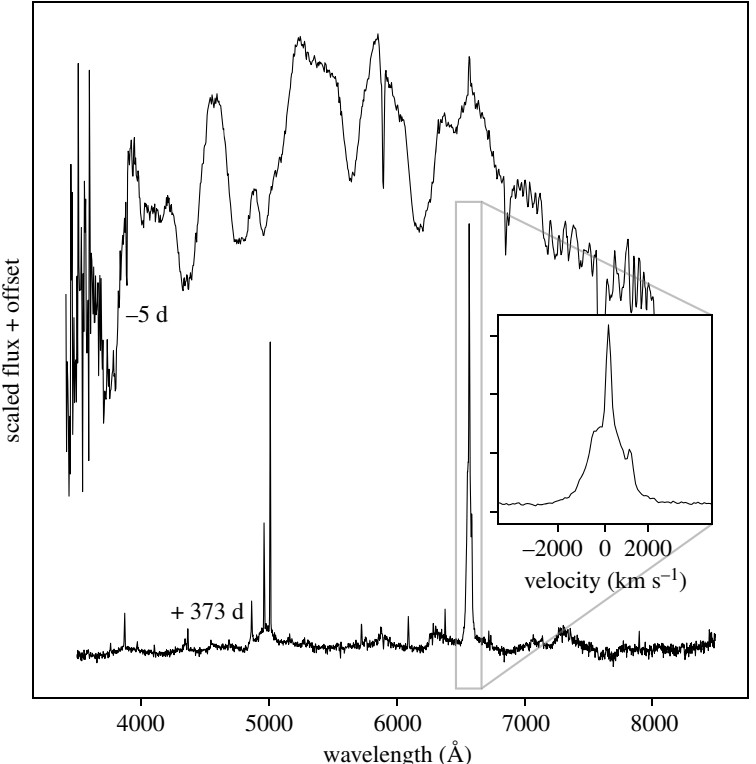

**Figure 4.** The pre-maximum classification spectrum for SN 2014C [137] showing it to be a fairly typical Type Ib SN, with broad photospheric features and no signs of H; also shown is a spectrum of SN 2014C one year later, revealing a metamorphosis to a Type IIn SN [135]. The inset panel shows H$\alpha$ in velocity space.

particularly appealing for some of these events, as they can potentially explain undulations in some late-time radio light curves. In this scenario, the wind from the progenitor and a binary companion can collide to create a CSM with a varying density profile, and a bumpy radio light curve [132]. Late-time optical observations in at least two Type IIb SNe also indicate the presence of a H-rich CSM, and it has been suggested that the absence of such spectral signatures in other cases may reflect different phases of strong mass transfer at the point of explosion [133].

One of the most surprising discoveries of recent years has been that some stripped envelope SNe that are *entirely* H free at early times appear to interact with H-rich CSM at late times [134–136]. Many of these SNe were originally classified as Type Ib events; with spectral signatures of He but no H in the SN ejecta. For a shell of H-rich CSM to be close enough to still interact with the SN ejecta implies that the mass loss occurred relatively soon before the star exploded; indeed in the case of SN 2001em this mass loss must have started only one or two thousand years before the SN explosion [134].

In figure 4 we show the dramatic spectral change for one such event, SN 2014C, where broad H-free photospheric SN features consistent with a typical Type Ib SN developed into those of a Type IIn [135,138]. In this case, late-time near and mid-infrared observations also allowed for the detection of dust in the circumstellar environment. Uniquely, in order to reproduce the infrared spectral energy distribution a mixture of dust compositions was required [139], as well as multiple CSM densities. This may imply that some of the CSM actually came from a binary companion to the star that exploded.

An even more extreme example can be seen for the Type Ic SN 2017ens [140]. Initially the spectra appear similar to a broad-lined Type Ic SN, indicative of an energetic explosion in a stripped star. At late times, the SN developed broad 2000 km s$^{-1}$ Balmer emission lines, together with high ionization lines that indicate a dense CSM and strong interaction. While there is clearly a wind component with a velocity of approximately 50 km s$^{-1}$, as measured from narrow absorption features, the 2000 km s$^{-1}$ material with which the ejecta interacts at late times is hard to explain. It is possible that this is the ejected former H envelope of the star, perhaps even lost through pulsational pair instability explosions [112].

As well as late-time interaction, some Type Ic (i.e. H- and He-poor) SNe have been seen to interact with H-rich CSM at early times (SN 2017dio; [141]). The mass loss rate required to explain the light curve of this

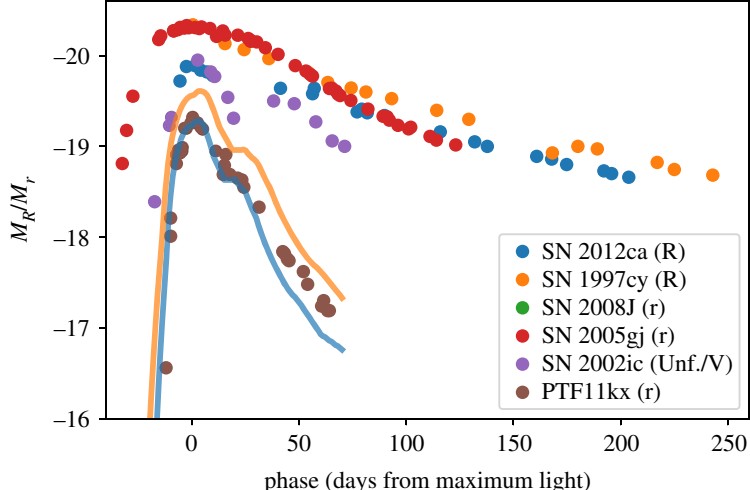

**Figure 5.** Light curves for purported Ia-CSM: PTF11kx [145], SN 2012ca [44], SN 1997cy [146] (as the maximum of SN 1997cy is not well constrained, it is plotted with respect to the discovery epoch), SN 2008J [147] (note that the extinction towards this SN is large and relatively uncertain), SN 2005gj [148], SN 2002ic [149]. In addition, template R-band light curves from [150] for a branch-normal Type Ia SN (blue line) and a SN 1991T-like event (orange line) are shown.

SN is exceptionally high, at approximately $0.02\,M_\odot\,yr^{-1}$. To sustain this over a number of decades would almost certainly require repeated eruptions, or possibly stripping by a binary companion.

Finally, it is quite possible that signatures of late-time circumstellar interaction have been missed in many stripped envelope SNe. Taking SN 2004dk as an example, strong narrow H$\alpha$ emission was first seen in a spectrum taken more than a decade after the SN explosion [142]. The radio light curve of SN 2004dk was seen to re-brighten around 5 years after explosion, suggesting the interaction started at this phase [143]. However, only a small handful of SNe have spectra or radio monitoring at such late times. In a H$\alpha$ imaging survey, Vinko *et al.* detected time-varying (to rule out a H II region) H$\alpha$ emission at the site of 15% of stripped envelope SNe [144]. However, this is likely to be an underestimate of the fraction of Type Ibc SNe with late-time interaction, as there were significantly more SNe in their sample where H$\alpha$ emission was detected, but no time variation was seen, including SN 2014C.

## 2.9. Ia-CSM

While the focus of this review has been core-collapse explosions of massive stars interacting with circumstellar material, there are also a number of examples in the literature of Type Ia SNe linked to circumstellar interaction (figure 5). This is perhaps unsurprising, as one of the two commonly discussed channels that may lead to a Type Ia SN sees a carbon–oxygen white dwarf accreting material from a non-degenerate binary companion (the single degenerate scenario; see [151,152] for reviews). Such a progenitor system may look similar to the galactic recurrent nova RS Ophiuchi, where ongoing accretion from a red giant onto a white dwarf will plausibly grow the latter to reach the Chandrasekhar mass on approximately Myr timescales [153]. Provided the mass loss rate from the companion is sufficiently high (greater than approx. $10^{-7}\,M_\odot\,yr^{-1}$), systems like RS Ophiuchi will develop a complex and structured circumstellar environment [154–156].

A number of apparently 'normal' Type Ia SNe have observed time-varying and blue-shifted NaD absorption, which has been linked to the outflowing wind of a companion [157–159]. While this may appear to be strong evidence for the single degenerate scenario, some models of double degenerate Type Ia SNe also predict the presence of approximately $10^{-5}\,M_\odot$ of CSM at velocities of $10$–$100\,km\,s^{-1}$ [160]. Another means of probing the presence of small amounts of CSM around Type Ia SNe is through searching for weak H$\alpha$ emission at late times. However, with few exceptions these searches have all yielded a null result [161–163].

In some instances, the signatures of circumstellar interaction are unambiguous, namely strong narrow H$\alpha$ emission that is clearly not associated with an underlying host galaxy. Perhaps the best example of this is PTF11kx, an unambiguous Type Ia SN that revealed signs of strong circumstellar interaction from around two months after explosion [145]. The light curve and spectra of PTF11kx are well explained by

collision of SN ejecta with greater than approximately $0.06\,M_\odot$ of CSM in shells of material that were ejected by a symbiotic nova progenitor [145,164].

What makes PTF11kx a relatively clear-cut case is that the SN displays the spectrum of a normal Type Ia before the interaction begins. For the other SNe that have been proposed as members of the class of 'Ia-CSM', the nature of the underlying SN is much less certain. A number of candidate SNe Ia-CSM are shown in figure 5, and it is clear that with the exception of PTF11kx, most of the purported SNe Ia-CSM are significantly brighter than Type Ia SNe, and have light curves that decline relatively slowly. Spectroscopically, these events appear relatively homogeneous, with narrow H$\alpha$, a distinctive pseudo-continuum around 5000 Å, and strong broad emission in the Ca NIR triplet. The debate in the literature as to the nature of these events is exacerbated by the fact that in most of these cases, the underlying SN is masked almost entirely by the CSM [165], making the task of inferring its nature much harder (see for example the various interpretations offered for the prototypical SN 2002ic [149,166,167]). However, one clue can come from the total radiated luminosity, which for some events may strain the energy budget of a Type Ia SN [44].

## 2.10. Impostors and non-supernovae

In addition to the classes of interacting core-collapse or thermonuclear SNe described previously, there is also evidence for interaction with CSM in a number of *non-terminal* transients. Perhaps the most striking example of this is in $\eta$ Car, as discussed earlier in this article, but there are also a number of examples of outbursts from massive stars both in our own galaxy (such as P-Cygni), and at extragalactic distances (often termed 'supernova impostors', e.g. [168,169]). In some cases, these may even presage the subsequent supernova explosion.

Along with outbursts from massive stars, there are a handful of so-called 'gap transients' associated with likely stellar mergers. Often termed 'luminous red novae', they display narrow H$\alpha$ associated with the material lost during the merger process and ejection of the common envelope. A detailed discussion of these transients is beyond the scope of this paper, and the reader is referred to the recent review in [170].

We also note that a number of peculiar 'fast blue optical transients' (FBOTs) have been recently reported in the literature, most notably AT 2018cow [171–173]. The nature of these events is still strongly contested, with suggestions ranging from failed (or otherwise peculiar) supernovae, through various scenarios involving jets, to the tidal disruption of a star by an intermediate mass black hole. At least some explanations proposed for AT 2018cow and other FBOTs such as CSS161010 invoke interaction with a dense CSM [172,174], or shock breakout from an optically thick wind [175].

## 2.11. SN 1987A

Finally, no survey of CSM and CSM interaction could be complete without touching briefly on SN 1987A. While SN 1987A is a quite unusual SN (estimates for the rate of 87A-like events are in the order of a few per cent [176]), its proximity has enabled a detailed study to be made of its circumstellar environment that is not possible for more distant events. Around 1 year after explosion, narrow emission lines were detected in the spectrum of SN 1987A and suggested to arise from a shell of CSM [177–179]. Subsequent observations with the Hubble Space Telescope (HST) confirmed the presence of a resolved ring of gas at a radius of approximately 0.2 pc from the SN [180]. It was suggested early on that this ring could be explained by emission from the shock region where the fast blue supergiant wind from the progenitor runs into the slower wind from its earlier red supergiant phase [177,178].

Since then, an ongoing effort with HST (summarized in [181]) has revealed the circumstellar environment in exquisite detail. In particular, continuing observations over three decades since SN 1987A have shown how the inner equatorial ring has been illuminated by X-rays from late-time interaction with the ejecta (figure 6), and helped constrain the mass loss history of the progenitor [182].

# 3. Conclusion

From the preceding, it should be clear that our study of circumstellar interaction in supernovae is still at an early stage. The ultimate goal of this effort, which is to explain individual transients and classes of transients with specific progenitors and physical scenarios, remains largely incomplete. Nonetheless, there are a number of key points that we can already be confident of:

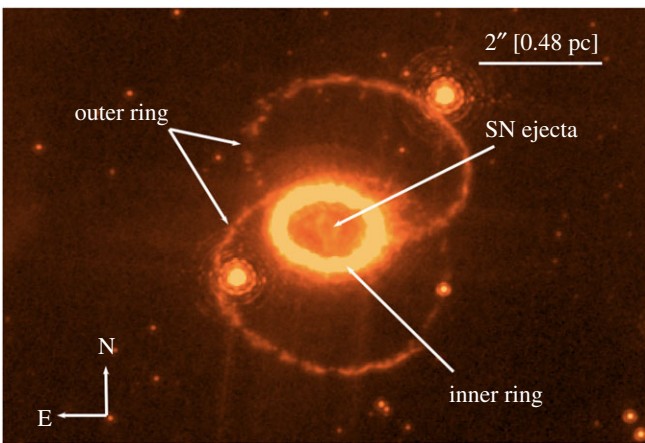

**Figure 6.** Narrow-band Hα (F658N) image of SN 1987A taken in December 2006 with the Hubble Space Telescope + ACS/HRC. The resolved SN ejecta, together with the inner and outer rings are indicated. The two bright point sources that are close to the outer rings are unrelated stars seen in projection.

— Interaction with CSM is not confined to a particular mass range or class of progenitors. Rather it can affect lightcurves and spectra for *all* types of SN, and progenitors of any mass.

— Many massive stars experience enhanced mass loss immediately prior to core collapse (on timescales of centuries to months).

— Mass loss is not confined to stellar winds, but rather can take the form of episodic eruptions and outbursts for stars across a range of masses.

— The circumstellar medium around supernovae is often highly non-uniform, with complex geometry such as rings or bipolar configurations.

— Some massive stars apparently explode as supernovae while in an LBV-like phase, contrary to the predictions of stellar evolutionary models.

Looking to the future, the major obstacle to furthering our understanding of CSM interaction and Type IIn SNe is not finding large numbers of transients. Rather, the limitation lies in spectroscopic classification and monitoring, which at present can only be obtained for a small fraction of events. With the advent of new instruments, and in particular high efficiency intermediate resolution spectrographs on 2 and 4 m class telescopes, this bottleneck will hopefully be alleviated. Along with observational developments, progress in theory and modelling are needed to firmly establish the mechanisms for pre-SN outbursts, and to make predictions as to the appearance of ejecta interacting with non-spherically symmetric CSM.

Data accessibility. Figures 2–5 show previously published data taken from the literature; references are given in the caption to each figure. Some light-curve data have been downloaded from the Open Supernova Catalog (https://sne.space/; [183]), while spectra have been obtained through WISeREP (https://wiserep.weizmann.ac.il; [184]). The HST image of SN 1987A shown in figure 6 was downloaded from the Hubble Legacy Archive (https://hla.stsci.edu/), while the colour images shown in figure 1 are available from the ESA HST webpages at www.spacetelescope.org/images/.

Competing interests. I declare I have no competing interest.

Funding. I gratefully acknowledge the support of a Royal Society – Science Foundation Ireland University Research Fellowship.

Acknowledgements. I am deeply grateful to the many collaborators with whom I have had the pleasure of working on the topic of circumstellar interaction. Especial thanks go to Andrea Pastorello, Rubina Kotak, Cosimo Inserra, Seppo Mattila, Stephen Smartt, Nancy Elias Rosa, Ting-Wan Chen and Anders Jerkstrand, who I have worked closely with on many of the supernovae discussed in this article. In addition, I thank my current and former students Emma Callis, Seán Brennan, Robert Byrne, Thomas Reynolds and Shane Moran for their ongoing work in this area. Finally, I thank the two anonymous referees for their careful reading and helpful suggestions.

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
