## [Reviewer comments · Royal Society Open Science]

Review History

RSOS-200467.R0 (Original submission)

Review form: Reviewer 1

Is the manuscript scientifically sound in its present form?

Yes

Are the interpretations and conclusions justified by the results?

Yes

Is the language acceptable?

Yes

Do you have any ethical concerns with this paper?

No

Have you any concerns about statistical analyses in this paper?

No

Recommendation?

Accept with minor revision (please list in comments)

Comments to the Author(s)

Dear Author,

This is a comprehensive review of CSM and CSM interactions in SNe and transients. I recommend it for acceptance with very minor revisions (see Appendix A).

"?" indicate questions or optional suggestions. I hope my hand writing is readable. Most comments are very small and straightforward.

Well done!

Review form: Reviewer 2

Is the manuscript scientifically sound in its present form?

Yes

Are the interpretations and conclusions justified by the results?

Yes

Is the language acceptable?

Yes

Do you have any ethical concerns with this paper?

No

Have you any concerns about statistical analyses in this paper?

No

Recommendation?

Major revision is needed (please make suggestions in comments)

Comments to the Author(s)

M. Fraser has provided a review of select transients and supernovae that interact with circumstellar environments. Phenomena ranging from Type Ia-CSM supernovae to diverse core collapse explosion scenarios are covered. The focus is on optical emission. Overall the paper is of good merit and will contribute to the literature cataloging the ever-growing taxonomy of supernova explosions. However, before I can recommend publication in Royal Society Open Science, I request that the author kindly consider my comments below.

--=[Major comments]=--

1) Section 1a titled "mass loss in massive stars" lacks proper discussion of the important influences that binary companions can have on massive star mass loss. Approximately 2/3 of stars will have binary companions close enough so that at some point interaction will affect mass loss (Sano et al. 2012; Moe & Di Stefano 2017). Although challenging to describe analytically, there are growing efforts at attempting to understand how companion stars drive mass loss in supernova progenitor systems. Important papers include: Podsiadlowski et al. 1992; Eldridge et al. 2008; Yoon et al. 2010,2017; Sravan et al. 2019. Stellar rotation is another important variable of in mass loss. See, e.g., Georgy et al. 2011,13; Groh et al. 2019. The associated timescales and geometries of these poorly understood mass loss mechanisms are critical to correctly interpret emission from SN-CSM interaction.

2) Section 2i titled "Ic with late time interaction" should be retitled "Ib and Ic with late time interaction." SN 2001em was of Type Ib (see Gal-Yam 2017) as was SN 2014C. The Type Ib event SN 2004dk should also be highlighted (Maurehan et al. 2018). It may be appropriate to acknowledge the work monitoring for this late-time interaction by Vinko et al. (2017).

3) There is little discussion about evolved supernovae visible > 5 yr after explosion due to interaction with CSM. SN 1980K can be considered the first supernova discovered by late-time interaction with CSM (Fesen & Becker 1988). See Milisavljevic et al. (2012) for a review and discussion of other interesting events; e.g., SN 1970G, SN 1993J. There are also events where the original supernova was not observed (and thus the classification is unclear) but they were discovered at late times due to strong SN-CSM interaction; e.g., SN 1986J (Rupen et al. 1987) and SN 1996cr (Bauer et al. 2008).

--=[Minor comments]=--

4) Consider including the recent multiwavelength review by Chandra (2018) in the list of references for section 1.

5) In general there is a lack of discussion of Type IIb events. See, e.g., Chevalier & Soderberg (2010). Type IIb supernovae may be associated with two progenitor pathways distinguished by mass loss leading up to core collapse (Maeda et al. 2015).

6) Recently, there has been a surge of interest in fast luminous transients; e.g., AT2018cow (Margutti et al. 2019; Perley et al. 2019), ZTF18abvkwla (Ho et al. 2020), and CSS161010 (Coppejans et al. 2020). Some interpretations of these objects include interaction with CSM; see previous citations (especially Coppejans et al. 2020) and, e.g., Fox et al. 2019.

7) Section 2h, "SNe 1994 and 2009kn" ==> "SNe 1994W and 2009kn"

8) Figure 1 caption "Carare" ==> "Car are"

9) All figures showcase early-phase interaction. Consider including an additional figure (with appropriate discussion) highlighting cases of late-phase interaction (e.g., SN 2014C with dramatic changes in spectral emissions).

Decision letter (RSOS-200467.R0)

18-May-2020

Dear Dr Fraser,

The editors assigned to your paper ("Supernovae and transients with circumstellar interaction") have now received comments from reviewers. We would like you to revise your paper in accordance with the referee and Associate Editor suggestions which can be found below (not including confidential reports to the Editor). Please note this decision does not guarantee eventual acceptance.

Please submit a copy of your revised paper before 10-Jun-2020. Please note that the revision deadline will expire at 00.00am on this date. If we do not hear from you within this time then it will be assumed that the paper has been withdrawn. In exceptional circumstances, extensions may be possible if agreed with the Editorial Office in advance. We do not allow multiple rounds of revision so we urge you to make every effort to fully address all of the comments at this stage. If deemed necessary by the Editors, your manuscript will be sent back to one or more of the

original reviewers for assessment. If the original reviewers are not available, we may invite new reviewers.

- Data accessibility

If you wish to submit your supporting data or code to Dryad (<http://datadryad.org/>), or modify your current submission to dryad, please use the following link:
<http://datadryad.org/submit?journalID=RSOS&manu=RSOS-200467>

- Competing interests

- Authors' contributions

AB carried out the molecular lab work, participated in data analysis, carried out sequence alignments, participated in the design of the study and drafted the manuscript; CD carried out

the statistical analyses; EF collected field data; GH conceived of the study, designed the study, coordinated the study and helped draft the manuscript. All authors gave final approval for publication.

- Acknowledgements

- Funding statement

on behalf of Dr Michelle Collins (Associate Editor) and Rob Ivison (Subject Editor)
openscience@royalsociety.org

Associate Editor's comments (Dr Michelle Collins):
Comments to the Author:

We have received two positive reviews on your manuscript. One requires some minor revisions, and the second a few more significant revisions. We encourage you to consider these revisions, and update your article accordingly before we accept this work.

Best wishes.
Reviewers' Comments to Author:
Reviewer: 1

Comments to the Author(s)
Dear Author,

This is a comprehensive review of CSM and CSM interactions in SNe and transients. I recommend it for acceptance with very minor revisions (see attached scan of annotated paper).

"?" indicate questions or optional suggestions. I hope my hand writing is readable. Most comments are very small and straightforward.

Well done!

Reviewer: 2

Comments to the Author(s)
M. Fraser has provided a review of select transients and supernovae that interact with circumstellar environments. Phenomena ranging from Type Ia-CSM supernovae to diverse core collapse explosion scenarios are covered. The focus is on optical emission. Overall the paper is of good merit and will contribute to the literature cataloging the ever-growing taxonomy of

supernova explosions. However, before I can recommend publication in Royal Society Open Science, I request that the author kindly consider my comments below.

--=[Major comments]=--

1) Section 1a titled "mass loss in massive stars" lacks proper discussion of the important influences that binary companions can have on massive star mass loss. Approximately 2/3 of stars will have binary companions close enough so that at some point interaction will affect mass loss (Sano et al. 2012; Moe & Di Stefano 2017). Although challenging to describe analytically, there are growing efforts at attempting to understand how companion stars drive mass loss in supernova progenitor systems. Important papers include: Podsiadlowski et al. 1992; Eldridge et al. 2008; Yoon et al. 2010,2017; Sravan et al. 2019. Stellar rotation is another important variable of in mass loss. See, e.g., Georgy et al. 2011,13; Groh et al. 2019. The associated timescales and geometries of these poorly understood mass loss mechanisms are critical to correctly interpret emission from SN-CSM interaction.

2) Section 2i titled "Ic with late time interaction" should be retitled "Ib and Ic with late time interaction." SN 2001em was of Type Ib (see Gal-Yam 2017) as was SN 2014C. The Type Ib event SN 2004dk should also be highlighted (Maurehan et al. 2018). It may be appropriate to acknowledge the work monitoring for this late-time interaction by Vinko et al. (2017).

3) There is little discussion about evolved supernovae visible > 5 yr after explosion due to interaction with CSM. SN 1980K can be considered the first supernova discovered by late-time interaction with CSM (Fesen & Becker 1988). See Milisavljevic et al. (2012) for a review and discussion of other interesting events; e.g., SN 1970G, SN 1993J. There are also events where the original supernova was not observed (and thus the classification is unclear) but they were discovered at late times due to strong SN-CSM interaction; e.g., SN 1986J (Rupen et al. 1987) and SN 1996cr (Bauer et al. 2008).

--=[Minor comments]=--

4) Consider including the recent multiwavelength review by Chandra (2018) in the list of references for section 1.

5) In general there is a lack of discussion of Type IIb events. See, e.g., Chevalier & Soderberg (2010). Type IIb supernovae may be associated with two progenitor pathways distinguished by mass loss leading up to core collapse (Maeda et al. 2015).

6) Recently, there has been a surge of interest in fast luminous transients; e.g., AT2018cow (Margutti et al. 2019; Perley et al. 2019), ZTF18abvkwla (Ho et al. 2020), and CSS161010 (Coppejans et al. 2020). Some interpretations of these objects include interaction with CSM; see previous citations (especially Coppejans et al. 2020) and, e.g., Fox et al. 2019.

7) Section 2h, "SNe 1994 and 2009kn" ==> "SNe 1994W and 2009kn"

8) Figure 1 caption "Carare" ==> "Car are"

9) All figures showcase early-phase interaction. Consider including an additional figure (with appropriate discussion) highlighting cases of late-phase interaction (e.g., SN 2014C with dramatic changes in spectral emissions).

Author's Response to Decision Letter for (RSOS-200467.R0)

See Appendix B.

Decision letter (RSOS-200467.R1)

16-Jun-2020

Dear Dr Fraser,

It is a pleasure to accept your manuscript entitled "Supernovae and transients with circumstellar interaction" in its current form for publication in Royal Society Open Science.

Kind regards,
Lianne Parkhouse
Royal Society Open Science
openscience@royalsociety.org

on behalf of Dr Michelle Collins (Associate Editor) and Rob Ivison (Subject Editor)
openscience@royalsociety.org

Appendix A

ROYAL SOCIETY OPEN SCIENCE

rsos.royalsocietypublishing.org

Article submitted to journal

Subject Areas:
xxxxx, xxxxx, xxxxx

Keywords:
xxxx, xxxxx, xxxxx

Author for correspondence:
Morgan Fraser
e-mail: morgan.fraser@ucd.ie

Supernovae and transients with circumstellar interaction

Morgan Fraser¹

¹School of Physics, O'Brien Centre for Science North,
University College Dublin, Belfield, Dublin 4, Ireland.

It is thirty years since the characteristic signatures of interaction with circumstellar material were first observed in a core-collapse supernova. Since then, CSM interaction has been observed and inferred across a range of transients, from the low-energy explosions of low-mass stars as likely electron-capture supernovae, through to the brightest superluminous supernovae. In this review, I present a brief overview of some of the interacting supernovae and transients that have been observed to date, and attempt to classify and group them together in a phenomenological framework.

(CSM)

1. Introduction and background

Wide-field synoptic surveys are discovering an ever growing number of transients, with nearly 20,000 publicly reported in 2019 alone. Of these transients, just under 2,000 were spectroscopically classified as supernovae, while the majority remained unclassified due to limited follow-up facilities. With such a large volume of transients it is unsurprising that rare, peculiar and unusual events are being found at an ever-increasing rate. Among these are many supernovae (SNe), and indeed other phenomena, which show signs of interaction with circumstellar material (CSM).

If a star explodes or erupts into a dense CSM, the faster moving ejecta will collide with slower moving circumstellar material. A forward shock is launched into the CSM, while a reverse shocks moves back into the expanding ejecta. These shocks will convert some of the kinetic energy of the ejecta into radiation; as the gas at the shock front is heated to $> 10^7$ K, this radiation will come in the form of ultraviolet and x-ray emission. The X-ray/UV flux will in turn ionize and excite gas surrounding the SN, giving rise to the narrow lines that are a classical signature of interaction. Along with this, non-thermal emission can arise from the acceleration of relativistic electrons at the shock (see [1,2] for comprehensive reviews of these processes).

capital X ?
b
X-ray

© 2014 The Authors. Published by the Royal Society under the terms of the Creative Commons Attribution License <http://creativecommons.org/licenses/by/4.0/>, which permits unrestricted use, provided the original author and source are credited.

references missing publisher? some repeated refs pp. range

THE ROYAL SOCIETY
PUBLISHING

many book refs are not complete

<https://mc.manuscriptcentral.com/rsos>

Figure 1. Left: Colour composite (filters F555W, F656N, F675W) made from Hubble Space Telescope images of WR 124. Right: Hubble Space Telescope colour composite for η Car, (filters F280N, F336W, F658N). Both WR 124 and η Car are examples of evolved massive stars with significant CSM; 1-2 M_{\odot} in the case of the former [3], and from ~ 10 to $\gtrsim 40 M_{\odot}$ for the latter [4,5]. Image credit: NASA/ESA/Schmidt.

rsos.royalsocietypublishing.org R. Soc. open sci. 0000000

Carinae! or *Carina*

(a) Mass loss in massive stars

Interacting supernovae and transients give an important insight into mass loss from massive stars in the final stages of their evolution. The evolution and ultimate fate of a massive ($> 8M_{\odot}$) star is, to a large extent, governed by the mass-loss it will experience over its lifetime. For a single star which is born with a Zero Age Main Sequence (ZAMS) mass of $\lesssim 30 M_{\odot}$, stellar winds will be insufficient to remove the H envelope, and if the star explodes as a core-collapse supernova it will appear as a H-rich Type II SN [6]. For stars more massive than this, stellar winds (at solar metallicity) are sufficient to remove all or part of the H or He envelope before the star explodes [7] (Fig. 1). In this case, the resulting supernova will be a Type Ib or Ic event, depending on whether only the H, or both the H and He envelopes are removed. *In many cases the mass of a star at the point of explosion will be less than half of its ZAMS mass.*

space missing?

20-30%? depending on Z

H → I

The stellar winds responsible for removing this mass are ultimately driven by the transfer of momentum from photons to the gas in the outer layers of the star. While it is generally accepted that this process will scale with both the luminosity of the star and its metallicity (as UV absorption lines are required to absorb photons), many of the details remain uncertain (see [8] for a comprehensive review). In particular, recent work has shown that winds are not smooth, but are instead clumpy, with local over- and under-densities of material that will affect how radiation is absorbed. Critically, clumping can change mass loss rates by an order of magnitude or more [9,10]. The choice of mass loss rate will have a strong effect on stellar models, in particular for stars $> 30 M_{\odot}$ [11].

Along with winds, stars can also lose mass through episodic eruptions. While the physical mechanism behind these eruptions is presently unknown, there are clear historical examples of them occurring, most famously for η Carinae. In the middle of the 19th century, η Carinae experienced a decade long outburst which saw it eject $\sim 10 M_{\odot}$, forming the spectacular Homunculus Nebula which can be observed today (Fig. 1) [4]. The so-called "Great Eruption" saw η Carinae for a period become one of the brightest stars in the sky. The inferred mass loss rate of η Carinae during this period (around $1 M_{\odot} \text{yr}^{-1}$) is much greater than can be accounted for by normal mass loss mechanisms, and requires either a Super-Eddington wind or some kind of explosive outburst.

One of the many remarkable properties of η Car is that some of the material lost during the Great Eruption had velocities of $\sim 10^4 \text{ km s}^{-1}$ [12]. This is much faster than the typical

right

wind velocity of even a Wolf-Rayet star, and points towards an explosive eruption. Several of the proposed explanations for what triggered this eruption invoke mass transfer in an eccentric binary or a stellar merger [13–16], motivated in part by the coincidence between the start of one of the subsequent outbursts of η Car and periastron [17]. Alternatively, instabilities in the later stages of nuclear burning, or perhaps associated with shell ignition could also provide the necessary energy. While many of these scenarios are plausible, the identification of ejecta at considerable distance from η Car consistent with outbursts as long as ~ 800 yr ago [18] pose a challenge.

(b) Interacting transients

The first suggestion that SNe with narrow lines in their spectra were a distinct class was made by Schlegel in 1990, who also proposed the “IIn” nomenclature [19]¹ Since then, the number of Type IIn SNe with reasonably comprehensive followup observations has risen, with probably around one hundred well studied events in the literature to date. In this review, I present an overview of the current landscape for transients which show some evidence for circumstellar interaction, with a particular focus on supernovae². Such a survey will necessarily be biased and incomplete, not least because many of the possible classes are a matter of ongoing debate in the literature.

2. Observational classes of interacting SNe

The field of SN classification and taxonomy is rapidly evolving as new classes of transients are identified. A clear illustration of this can be seen in the review of Type Ia SNe by Taubenberger, where the prototypes of many of subclasses have only been identified in the last decade [23].

incomplete REF info.

Compounding the confusion in transient classes is that the taxonomic classification of SNe proceeds on an observational rather than physical basis. This is particularly challenging for interacting SNe, where CSM interaction may mask the underlying transient (for example, the debate over so-called “Type Ia-CSM” SNe has shown how a CSM-enshrouded thermonuclear explosion of a white dwarf can look remarkably similar to the core-collapse of a stripped star [24,25])

However, while the spectroscopic properties of Type IIn SNe often appear similar, they can display heterogeneous lightcurves (see, for example, the sample of spectroscopically confirmed SNe IIn with lightcurves from PTF [26]). Lightcurves can provide a better indicator of total energetics in an interacting supernova (especially as fast ejecta is usually masked behind CSM), while the shape and duration of the lightcurve can provide some indication of the mass and radial distribution of CSM.

of?

(a) Luminous, long lived Type IIn SNe

remove space

A number of Type IIn SNe display slow evolving, long lived lightcurves [27,28], and are commonly believed to be associated with the explosion of a very massive star into a massive (on the order of a few, to a few tens of M_{\odot}) CSM. The archetype of this group is SN 2010jl, where combined x-ray, radio and optical observations allowed for detailed modeling of the CSM density [29–31]. Various authors set a lower limit to the total CSM mass of $\times 3$ or $\times 10 M_{\odot}$, the exact limit depends to a large extent on the assumed wind velocity. Whatever the precise value, the radius of the CSM ($\sim 10^{16}$ cm [29]) is consistent with a relatively recent episode of enhanced mass loss, starting around 40 years prior. It is difficult to imagine anything other than a massive, LBV-like progenitor (perhaps even an extra-galactic η Car analog) giving rise to such a CSM configuration and SN. This would also be consistent with the location of SN 2010jl in a UV-bright highly starforming environment.

¹Although see Zwicky’s peculiar “Type V” SN 1961V as an example of an prior transient that is clearly affected by CSM interaction [20,21].
²See [22] for another recent review of this topic.

to the explosion?

Unfortunately SN 2010jl was first discovered when it was emerging from behind the Sun, and so there are only lower limits to the rise time to peak luminosity. However, a number of other long-lived and energetic Type IIn SNe (for example, PTF12glz [32] and HSC16aayt [33]) have shown a very slow and long rise time. This could be interpreted as being indicative of a long diffusion time for photons to escape from a massive optically thick CSM. Alternatively the long rise time could be due to an asymmetric CSM [32], possibly in the form of a circumstellar disk [34]. This latter scenario would be consistent with spectropolarimetry of SN 2010jl, which revealed substantial asymmetry [35].

type IIP
not
defined
when first
used

core collapses

(b) SN1998S-like events, SNe IIL and early interaction in Type IIP SNe

Non-interacting H-rich core collapses with a linearly declining lightcurve are designated "Type IIL" events [36]. While there is some debate as to whether Type IIL SNe are a discrete class, or whether they form a continuum with more common Type IIP events, they are generally accepted to arise from the explosion of a star that has too little H in its envelope to sustain a recombination powered plateau, but which are not so stripped as to give rise to a Type IIn SN. Interestingly, there are a number of apparent connections between Type IIL SNe and Type IIn events.

On the basis of their lightcurves alone, a number of purported Type IIn SNe such as SN 1998S [37] could be classified as Type IIL events. This SN experienced a linear decline, while the signatures of interaction, viz. strong narrow emission lines, only persisted for the first month after explosion [38]. It is worth noting that if SN 1998S had not been spectroscopically classified until ~1 month after explosion it would have been regarded as a Type IIL SN. A handful of similar events such as SN2013fc have been found to have a similar photometric evolution [39].

Even among unambiguous Type Type IIL SNe that show no obvious signs of narrow lines, it has been suggested that CSM interaction may still provide part of the luminosity (e.g. [40]). In general, radiation hydrodynamic models struggle to reproduce a bright linear lightcurve with no large drop onto a radioactive tail phase. One solution to this is to invoke an extremely inflated envelope (or superwind) for the progenitor, as suggested for the Type IIL SN 1979C [41]. Similarly, more recent modelling has shown that the lightcurves of Type IIL SNe can be best reproduced with an additional dense, confined CSM [42,43]. Furthermore, even some apparently normal Type IIP SNe have been found to show signs of CSM interaction at early times, requiring some additional material close to the progenitor lost just before explosion [44-46]. It is worth noting that x-ray observations at late times point towards a relatively high mass loss rate ($10^{-4} M_{\odot} \text{yr}^{-1}$) for the progenitor of SN 1979C [47], while fluctuations in the radio lightcurve are suggestive of an inhomogeneous CSM [48].

no h

(c) Superluminous and peculiar interacting SNe

Recent years have seen the discovery of increasing numbers of superluminous supernovae (SLSNe), with peak luminosities far in excess of what can plausibly be powered by radioactive decay [49]. Many of these SNe are H-poor, and are probably powered by some sort of additional central engine, such as a rapidly rotating magnetar which injects energy as it spins down [50]. A handful of these superluminous events are spectroscopically Type IIn SNe, where CSM interaction is presumably sufficient to boost the luminosity to brighter than $M_V \sim -22$.

The prototype of this class is SN 2006gy, which reached a peak magnitude of -22 , while displaying relatively narrow lines emission lines of H [51,52]. Modeling of the lightcurves and spectra are consistent with the collision of $\sim 10 M_{\odot}$ of ejecta with a similar mass of patchy CSM [52,53]. While these values are large, they are not inconceivable, and the fact that SN 2006gy was so luminous is simply a reminder that the conversion of kinetic energy to radiation at a shock can be a highly efficient process.

spaces
missing
10-170-04

Late time observations of H-poor SLSNe have revealed that some show strong spectroscopic signatures of interaction with H-rich CSM at late times (>6 months from explosion). What is remarkable is that the velocity of the Balmer emission lines that appear at such late times are

$\sim 5000 \text{ km s}^{-1}$. These velocities are much higher than the fastest stellar winds, and even the velocity of the bulk of the ejecta from typical LBV giant eruptions. In fact, the most plausible mechanism to eject a massive H envelope with this velocity is a pulsational pair instability explosion [54]. Additional support for this scenario comes from observations of the SLSN iPTF16eh, where a light echo seen in spectra is consistent with a massive circumstellar shell ejected thirty years before the SN [55].

Interaction has also been proposed as a power source for some peculiar H-rich SNe. While iPTF14hls only reached an absolute magnitude $r \sim -19$ at its peak, it remained bright for over two years, with multiple peaks in its lightcurve [56,57]. Remarkably, iPTF14hls displayed broad $>6000 \text{ km s}^{-1}$ H lines for the same duration, with very little evolution in line strength or velocity. Arcavi et al. suggested that the line-forming region was detached from the photosphere, and could be coming from a massive shell of some tens of M_{\odot} that was ejected by the progenitor a few years prior. This scenario was further strengthened by the tentative detection of a precursor outburst on photographic plates from the 1950's [56]. Remarkably, [58] presented a spectrum at >3 years from discovery which showed H α to have a double peaked structure, with two emission components at -1000 and $+700 \text{ km s}^{-1}$. Andrews and Smith [58] suggested a scenario where CSM was present in a disk around iPTF14hls, but that interaction was masked by fast ejecta that expanded out past the disk, blocking spectroscopic signatures of narrow lines for some viewing angles.

(d) Ibn

While the majority of interacting transients arise from the collision of ejecta with H-rich CSM, in a handful of cases the CSM appears to be He-rich. The archetype of this class is SN 2006jc [59,60], although there were some earlier identifications of narrow He lines in SN spectra, such as SN1999cq [61]. Since then, about thirty so-called "Ibn" SNe have been identified.

Photometrically, Type Ibn SNe are typically characterised by a rapid decline from a relatively bright peak, consistent with a small ejecta mass. Their spectra however, tend to be more varied [62], and in particular, there appears to be a continuum of H and He line strengths, where some events have much stronger He lines, while others appear to be intermediate between Types Ibn and IIn [63].

The progenitors of Type Ibn SNe have been suggested to be Wolf-Rayet stars (e.g. [64,65]), and this is appealing for a number of reasons. In particular, Wolf-Rayet stars are H-poor; while they have wind velocities comparable to the $\sim 10^3 \text{ km s}^{-1}$ seen in the narrow lines of Type Ibn SNe. However, it is worth noting that the fast decline of Ibn lightcurves in some cases points to a smaller ejecta mass than that of normal Type Ibc SNe. This is surprising, as the majority of the former are believed to come from lower mass stars that have been stripped in binaries. The progenitors of most Ibc SNe should hence have a smaller core mass than a WR star at the point of explosion, and hence one would expect WR stars to have broader lightcurves than Type Ibc SNe.

At least one Type Ibn SN displayed a pre-SN outburst around two years prior to explosion [59]. The nature of this outburst is still debated, although as SN 2006jc was likely in a binary system [66] it may well come from an LBV-like companion rather than the progenitor itself.

(e) SN2009ip-like events

One of the most intensively studied (and contentious) interacting transients is SN 2009ip. First discovered in 2009 and assigned a "SN" designation, SN 2009ip went on to display three years of erratic variability. During this phase the spectrum of SN 2009ip was hot and dominated by narrow CSM lines, reminiscent of an LBV eruption. Remarkably, even during this period spectra revealed some material moving at $\gtrsim 12,000 \text{ km s}^{-1}$ (from the blue edge of the H α absorption) [67,68]. Such velocities are indicative of an energetic and explosive event, and an associated blast wave which was even faster than that seen in η Car [69].

The erratic variability of SN 2009ip reached a dramatic conclusion in 2012, when the transient first brightened to around $M_R \sim -15$ over one month ("event A"), before reaching $M_R \sim -18$ in a

not clear

not 100% clear to me
 Note: maybe more massive WRs have weaker explosions
 -> Blts

missing comma here

second brightening event ("event B"). As no non-terminal eruption of a massive star had hitherto been observed to reach this luminosity, it was suggested that the 2012 B event was the final explosion of SN 2009ip as a (genuine) supernova [70]. Nonetheless, some authors have argued that it may not be a terminal core-collapse or remained agnostic (e.g. [68,71,72]), on the basis of the restrictive upper limit to the ejected ^{56}Ni mass and the absence of spectral features associated with SN nucleosynthesis.

If the "event B" of SN 2009ip was indeed a genuine SN, then we can consider the implications of the pre-SN outbursts for stellar evolutionary models. Any outburst immediately prior to core-collapse is plausibly triggered by one of the later phases of nuclear burning. The timescale for core Si burning is on the order of a day, while even in a $60M_{\odot}$ star core C burning commences about 50 yr before core-collapse [73]. Intermediate between these burning stages are core O and Ne ignition. [74] suggested that the progenitor of SN 2015bh (another member of the SN 2009ip-like class) had been in a state of variability for at least two decades before its 2015 outburst. It is important to note that these timescales come from one-dimensional models of stellar evolution. While these models well reproduce observables such as the locus of massive stars on a Hertzsprung-Russell diagram, they are necessarily less tested when it comes to the final stages of stellar evolution. Furthermore, there is a possibility that convective or turbulent instabilities that are seen in 3D models can drive outbursts [75,76]

include authors' name here?

One of the most intriguing aspects of SN 2009ip is that there are a surprising number of similar transients that have been found in the last few years [74,77–80]. These transients are almost identical in their spectra, have a similar bright peak and decline in their lightcurve, and share a pre-peak outburst. What makes this surprising is that SN 2009ip is a complex transient with a highly structured CSM. Multiple emission and absorption components are needed to fit the line profiles in SN 2009ip (e.g. [81]) pointing towards a complex geometry. Further evidence for this comes from spectropolarimetry [82,83], where the polarisation was interpreted as implying a disk-like geometry. From this, one would naturally expect an observationally diverse set of transients, thanks to differences in CSM, mass loss history, and critically, viewing angle effects.

The true nature of the SN2009ip-like transients remains uncertain. The steady decline and lack of new outbursts since 2012 points towards a terminal event. On the other hand, the transient has not yet faded significantly below the progenitor magnitude. Whether these events are core-collapse supernovae, extreme non-terminal eruptions, or even a star entering the Wolf Rayet phase is an open question [74].

(f) Ia-CSM

While the focus of this review has been core-collapse explosions of massive stars interacting with circumstellar material, there are also a number of examples in the literature of Type Ia SNe linked to circumstellar interaction (Fig. 2). This is perhaps unsurprising, as one of the two commonly discussed channels that may lead to a Type Ia SN sees a carbon-oxygen white dwarf accreting material from a non-degenerate binary companion (the single degenerate scenario; see [84,85] for (recent) reviews). Such a progenitor system may look similar to the galactic recurrent nova RS Ophiuchi, where ongoing accretion from a red giant onto a white dwarf will plausibly grow the latter to reach the Chandrasekhar mass on $\sim\text{Myr}$ timescales [86]. Provided the mass loss rate from the companion is sufficiently high ($\gtrsim 10^{-7} M_{\odot} \text{ yr}^{-1}$), systems like RS Ophiuchi will develop a complex and structured circumstellar environment [87–89].

A number of apparently "normal" Type Ia SNe have observed time-varying and blue-shifted NaD absorption, which has been linked to the outflowing wind of a companion [96–98]. While this may appear to be strong evidence for the single degenerate scenario, some models of double degenerate Type Ia SNe also predict the presence of $\sim 10^{-5} M_{\odot}$ of CSM at velocities of 10–100 km s^{-1} [99]. Another means of probing the presence of small amounts of CSM around Type Ia SNe is through searching for weak $\text{H}\alpha$ emission at late times. However, with few exceptions these searches have all yielded a null result [100–102].

Figure 2. Lightcurves for purported Ia-CSM: PTF11kx [90], SN2012ca [25], SN1997cy [91] (as the maximum of SN 1997cy is not well constrained, it is plotted with respect to the discovery epoch), SN2008J [92] (note that the extinction towards this SN is large and relatively uncertain), SN2005gj [93], SN2002ic [94]. In addition, template R-band lightcurves from [95] for a Branch-normal Type Ia SN (blue line) and a SN1991T-like event are shown.

In some instances, the signatures of circumstellar interaction are unambiguous, namely strong narrow H α emission that is clearly not associated with an underlying host galaxy. Perhaps the best example of this is PTF11kx, an unambiguous Type Ia SN that revealed signs of strong circumstellar interaction from around two months after explosion [90]. The lightcurve and spectra of PTF11kx are well explained by collision of SN ejecta with $\gtrsim 0.06M_{\odot}$ of CSM in shells of material that were ejected by a symbiotic nova progenitor [90,103].

What makes PTF11kx a relatively clear-cut case is that the SN displays the spectrum of a normal Type Ia before the interaction begins. For the other SNe that have been proposed as members of the class of "Ia-CSM", the nature of the underlying SN is much less certain. A number of candidate SNe Ia-CSM are shown in Fig. 2, and it is clear that with the exception of PTF11kx, most of the purported SNe Ia-CSM are significantly brighter than Type Ia SNe, and have lightcurves that decline relatively slowly. Spectroscopically, these events appear relatively homogeneous, with narrow H α , a distinctive pseudo-continuum around $\sim 5000 \text{ \AA}$, and strong broad emission in the Ca NIR triplet. It appears that in all these some cases, the underlying SN is masked entirely by the CSM [104], making the task of inferring its nature much harder. However, one clue can come from the total radiated luminosity, which for some events may strain the energy budget of a Type Ia SN [25].

(g) SN 2008S-like

The class of SN2008S-like IIn (also commonly referred to as Intermediate-Luminosity Red Transients, ILRTs) are characterised by faint absolute magnitudes ($-12.5 > M_R > -15$), a smoothly evolving lightcurve that in some cases appears to settle onto a faint tail, and a small ejected Ni mass (Fig. 3). Their spectra generally become quite red, and are dominated by narrow emission lines of H, Ca and Fe. In particular, the [Ca II] feature at $\lambda\lambda 7291, 7323$ is a characteristic signature of ILRTs³.

The prototype of this class was SN 2008S [107], which was suggested to result from a weak electron capture supernova [111] in a super-asymptotic giant branch progenitor. The progenitor was enshrouded in a dusty circumstellar medium, as it was detected at mid-infrared wavelengths

³ Although some objects continue to blur the apparent boundaries between ILRTs and other classes [105,106]

Figure 3. Lightcurves for SN 2008S-like transients [107–110]. The black line shows the expected decline rate from the decay of ^{56}Co .

with the Spitzer Space Telescope, but not at optical or NIR wavelengths. Similar conclusions have been drawn for other members of the class [112].

Alternative explanations for some of the SN 2008S-like events as non terminal outbursts have also been proposed, in particular that some of these may be evolved stars moving blue-ward on the Hertzsprung Russell diagram, or a high-mass x-ray binary [113,114]. A key test for this is whether there is a surviving star after the transient has faded, which remains a point of some debate in the literature [114,115].

(h) IIn-P

The term “IIn-P” SNe was first suggested by Mauerhan [116], who grouped together SNe 1994W, 2009kn and 2011ht. These events all follow a similar photometric evolution, with a relatively luminous plateau lasting around one hundred days, followed by a precipitous drop onto a slowly declining tail phase (Fig. 4). This evolution is at least superficially similar to that seen in Type IIP supernovae.

The ejected Ni mass for this class tends to be low, with estimates for SN1994W ranging from $\sim 10^{-2}$ to $\sim 10^{-3} M_{\odot}$ [117]; while SN 2009kn had a ejected Ni mass of $\sim 0.02 M_{\odot}$ [118]. As noted by [117], attempts to estimate or even place limits on the Ni mass for these SNe are somewhat fraught. A late time NIR excess (as seen in SN 2009kn) is suggestive of dust formation, implying that we may underestimate the tail luminosity. Equally, the rapid decline seen in SN 1994W, which is faster than the $0.98 \text{ mag } 100\text{d}^{-1}$ decline rate expected from ^{56}Ni , implies that γ -rays are not fully trapped in the ejecta, and hence some of the energy from radioactive decay may be leaking out unobserved. On the other hand, spectral signatures of CSM interaction persist to these late phases. If the SN is not entirely powered by radioactive decay, measurements of the tail phase luminosity will overestimate the Ni mass (through γ). These caveats notwithstanding, unless some material is lost through fallback onto the compact remnant, low ejected Ni masses are broadly consistent with a low energy explosion of a star at the lower extremum of the mass range for core-collapse [116,119].

The spectra of IIn-P are characterised at early times by a hot, blue continuum with narrow Balmer emission lines. The Balmer lines also display a P-Cygni absorption component with a velocity minimum around $400 - 800 \text{ km s}^{-1}$ [118,121]. As the spectrum cools, a forest of narrow P-Cygni FeII and TiII lines are revealed, together with both forbidden and permitted transitions

Fig. 4?
move
up
to next
page

of Ca. Broad ($\sim 10^4 \text{ km s}^{-1}$) ejecta features are not seen at any phase, and while [O I] lines are observed in SN 2011ht at late phases [116], these are extremely weak.

Interestingly, a precursor outburst was seen around 6 months prior to SN 2011ht [120]. While this is plausibly associated with the ejection of material from the progenitor, it does not allow us to infer whether SN 2011ht itself was a terminal and non-terminal event. In the case of a core-collapse SN, the precursor outburst may be driven by instabilities in the later stages of nuclear burning [120]. Similar outbursts can not be ruled out for SNe 1994 and 2009kn, as they lack pre-discovery imaging.

While most authors have argued that SN 2011ht was a core-collapse explosion [116,118,123], it is worth noting that some have suggested that SN 1994W and SN 2011ht were actually non-terminal outbursts [121,124]. Humphreys et al. argued that a non-terminal event could equally explain the observations of 2011ht, and that while a mass loss rate of $\sim 0.05 M_{\odot} \text{ yr}^{-1}$ would be required, this is potentially consistent with a super-Eddington giant eruption from a massive star. In the case of SN 1994W, Dessart et al. proposed a scenario where successive shells of material ejected by the progenitor could collide, leading to a luminous transient. An alternative suggestion, based in large part on a strong spectral similarity between SN2011ht and the luminous red nova NGC4490-2011OT1, was that the outburst six months prior to SN 2011ht was in fact the ejection of a common envelope following a binary merger [125]. In this scenario, either the merger could have triggered a supernova explosion, or alternatively, 2011ht was caused by the collision of ejected shells.

(i) Ic with late time interaction

One of the most surprising discoveries of recent years has been that some stripped envelope SNe appear to interact with H-rich CSM at late times [126–128]. For a shell of H-rich CSM to be close enough to still interact with the SN ejecta implies that the mass loss occurred relatively soon before the star exploded; indeed in the case of SN 2001em this mass loss must have started only one or two thousand years before the SN explosion.

For one such event, SN 2014C, late time near and mid-infrared observations allowed the detection of dust in the circumstellar environment. Uniquely, in order to reproduce the infrared spectral energy distribution a mixture of dust compositions was required [129], as well as multiple CSM densities. This may imply that some of the CSM actually came from a binary companion to the star that exploded.

A more extreme example can be seen for SN 2017ens [130]. Initially the spectra appear similar to a broad-lined Type Ic SN, indicative of an energetic explosion in a stripped star. At late times, the SN developed broad 2000 km s^{-1} Balmer emission lines, together with high ionization lines that indicate a dense CSM and strong interaction. While there is clearly a wind component with a velocity of $\sim 50 \text{ km s}^{-1}$, as measured from narrow absorption features, the 2000 km s^{-1} material with which the ejecta interacts at late times is hard to explain. It is possible that (this the ejected former H envelope of the star, perhaps even lost through pulsational pair instability explosions [54].

As well as late time interaction, some Type Ic (i.e. H and He-poor) SNe have been seen to interact with H-rich CSM at early times (SN 2017dio; [131]). The mass loss rate required to explain the lightcurve of this SN is exceptionally high, at $\sim 0.02 M_{\odot} \text{ yr}^{-1}$. To sustain this over a number of decades would almost certainly require repeated eruptions, or possibly stripping by a binary companion.

(j) Impostors and non supernovae

In addition to the classes of interacting core-collapse or thermonuclear SNe described previously, there is also evidence for interaction with CSM in a number of *non-terminal* transients. Perhaps the most striking example of this is in η Car, as discussed earlier in this article, but there are also a number of examples of outbursts from massive stars both in our own galaxy (such as P-Cygni),

prologue?

and at extragalactic distances (often termed “supernova impostors” (e.g. [132,133]). In some cases, these may even prologue the subsequent supernova explosion.

Along with outbursts from massive stars, there are a handful of so-called “gap transients” associated with likely stellar mergers. Often termed “luminous red novae”, they display narrow H α associated with the material lost during the merger process and ejection of the common envelope.

A detailed discussion of these transients is beyond the scope of this paper, and the reader is referred to the recent review in [134].

(k) SN1987A

Finally, no survey of CSM and CSM interaction could be complete without touching briefly on SN 1987A. While SN 1987A is a quite unusual SN (estimates for the rate of 87A-like events are on the order of a few percent [135]), its proximity has enabled a detailed study to be made of its circumstellar environment that is not possible for more distant events. Around one year after explosion, narrow emission lines were detected in the spectrum of SN1987A and suggested to arise from a shell of CSM [136–138]. Subsequent observations with the Hubble Space Telescope confirmed the presence of a resolved ring of gas at a radius of ~ 0.2 pc from the SN [139]. It was suggested early on that this ring could be explained by emission from the shock region where the fast blue supergiant wind from the progenitor runs into the slower wind from its earlier red supergiant phase [136,137].

Since then, an ongoing effort with HST (summarised in [140]) has revealed the circumstellar environment in exquisite detail. In particular, continuing observations over three decades since SN 1987A have shown how the inner equatorial ring has been illuminated by x-rays from late time interaction with the ejecta (Fig. 5), and helped constrain the mass loss history of the progenitor [141].

3. Conclusion

From the preceding, it should be clear that our study of circumstellar interaction in supernovae is still at an early stage. The ultimate goal of this project, which is to explain individual transients and classes of transients with specific progenitors and physical scenarios, remains largely incomplete. Nonetheless, there are a number of key points that we can already be confident of:

not defined (not needed)

- Interaction with CSM is not confined to a particular mass range or class of progenitor. Rather it can affect lightcurves and spectra for *all* types of SN, and progenitors with any (ZAMS) mass.
- Many massive stars experience enhanced mass loss immediately prior to core-collapse (on timescales of centuries to months).
- Mass loss is not confined to stellar winds, but rather can take the form of episodic eruptions and outbursts for stars across a range of masses.
- The circumstellar medium around supernovae is often highly non-uniform, with complex geometry such as rings or bipolar configurations.
- Some massive stars explode as supernovae while in an LBV-like phase, contrary to the predictions of stellar evolutionary models.

Looking to the future, the major obstacle to furthering our understanding of CSM interaction and Type II_n SNe is not finding large numbers of transients. Rather, the limitation lies in spectroscopic classification and monitoring, which at present can only be obtained for a small fraction of events. With the advent of new instruments, and in particular high efficiency intermediate resolution spectrographs on 2- and 4-m class telescopes, this bottleneck will hopefully be alleviated. Along with observational developments, progress in theory and modeling are needed to firmly establish the mechanisms for pre-SN outbursts, and to make predictions as to the appearance of ejecta interacting with non-spherically symmetric CSM.

Appendix B

=====

In the first referee report (sent as a scanned PDF), there were a number of minor comments. All typos and minor changes suggested by the referee have been made with one exception. There was a comment that references to comparatively recently discovered classes of Type Ia SNe were missing. Since I now make clear that I am citing a review ("a clear illustration of this can be seen in the review of Type Ia SNe by Taubenberger, where the prototypes of many of subclasses...") and I am only using this as an example of the explosion in new transients found since the advent of modern surveys, I do not feel that there is a need to reference all the discovery papers.

=====

I address the comments by the second referee below:

--=[Major comments]=--

1) Section 1a titled "mass loss in massive stars" lacks proper discussion of the important influences that binary companions can have on massive star mass loss. Approximately 2/3 of stars will have binary companions close enough so that at some point interaction will affect mass loss (Sano et al. 2012; Moe & Di Stefano 2017). Although challenging to describe analytically, there are growing efforts at attempting to understand how companion stars drive mass loss in supernova progenitor systems. Important papers include: Podsiadlowski et al. 1992; Eldridge et al. 2008; Yoon et al. 2010,2017; Sravan et al. 2019. Stellar rotation is another important variable of in mass loss. See, e.g., Georgy et al. 2011,13; Groh et al. 2019. The associated timescales and geometries of these poorly understood mass loss mechanisms are critical to correctly interpret emission from SN-CSM interaction.

****DONE**** References and two paragraphs on binary mass loss and CSM added (end of Sect. 1a). In addition in Sect. 2h (Stripped envelope SNe with late time interaction) I have added a paragraph on the role of binarity in Type IIb SNe.

2) Section 2i titled "Ic with late time interaction" should be retitled "Ib and Ic with late time interaction." SN 2001em was of Type Ib (see Gal-Yam 2017) as was SN 2014C. The Type Ib event SN 2004dk should also be highlighted (Maurehan et al. 2018). It may be appropriate to acknowledge the work monitoring for this late-time interaction by Vinko et al. (2017).

****DONE**** New paragraph added, also changed subsection title to "Stripped envelope SNe with late time interaction" as we also discuss Type IIb SNe here.

3) There is little discussion about evolved supernovae visible > 5 yr after explosion due to interaction with CSM. SN 1980K can be considered the first supernova discovered by late-time interaction with CSM (Fesen & Becker 1988). See Milisavljevic et al. (2012) for a review and discussion of other interesting events; e.g., SN 1970G, SN 1993J. There are also events where

the original supernova was not observed (and thus the classification is unclear) but they were discovered at late times due to strong SN-CSM interaction; e.g., SN 1986J (Rupen et al. 1987) and SN 1996cr (Bauer et al. 2008).

****DONE**** I now mention this in Sect. 1b, but do not go into this in detail as discussed in the text.

--=[Minor comments]=--

4) Consider including the recent multiwavelength review by Chandra (2018) in the list of references for section 1.

****DONE****

5) In general there is a lack of discussion of Type IIb events. See, e.g., Chevalier & Soderberg (2010). Type IIb supernovae may be associated with two progenitor pathways distinguished by mass loss leading up to core collapse (Maeda et al. 2015).

****DONE**** Added in to Sect 2h

6) Recently, there has been a surge of interest in fast luminous transients; e.g., AT2018cow (Margutti et al. 2019; Perley et al. 2019), ZTF18abvkwla (Ho et al. 2020), and CSS161010 (Coppejans et al. 2020). Some interpretations of these objects include interaction with CSM; see previous citations (especially Coppejans et al. 2020) and, e.g., Fox et al. 2019.

****DONE**** I now mention these events in a paragraph in Sect. 2j. I do not go into these in much detail, as they are so recent that there is not a clear enough picture of what the FBOTs actually are, and anything I write will probably be out of date in a matter of months!

7) Section 2h, "SNe 1994 and 2009kn" ==> "SNe 1994W and 2009kn"

****DONE****

8) Figure 1 caption "Carare" ==> "Car are"

****DONE****

9) All figures showcase early-phase interaction. Consider including an additional figure (with appropriate discussion) highlighting cases of late-phase interaction (e.g., SN 2014C with dramatic changes in spectral emissions).

****DONE**** New Fig. 4

=====

In addition, I have made some minor clarifications and fixed typos throughout following comments from other readers. I have also added data sources section as requested.